# The Role of the lncRNA MALAT1 in Neuroprotection against Hypoxic/Ischemic Injury

**DOI:** 10.3390/biom12010146

**Published:** 2022-01-17

**Authors:** Liping Wang, Sijie Li, Sara Saymuah Stone, Na Liu, Kerui Gong, Changhong Ren, Kai Sun, Chunyang Zhang, Guo Shao

**Affiliations:** 1Center for Translational Medicine, The Third People’s Hospital of Longgang District, Shenzhen 518112, China; wlp19980317@163.com (L.W.); liu_na_china@163.com (N.L.); 2Inner Mongolia Key Laboratory of Hypoxic Translational Medicine, Baotou Medical College, Baotou 014060, China; 3Institute for Neuroscience, Baotou Medical College, Baotou 014060, China; 4Department of Emergency, Xuanwu Hospital, Capital Medical University, Beijing 100053, China; lisijie@xwh.ccmu.edu.cn; 5Beijing Institute of Brain Disorders, Capital Medical University, Beijing 100069, China; 6Department of Neurosurgery, Wayne State University School of Medicine, Detroit, MI 48021, USA; sara.saymuah2@med.wayne.edu; 7Department of Oral and Maxillofacial Surgery, University of California San Francisco, San Francisco, CA 94143, USA; keruigong@gmail.com; 8Beijing Key Laboratory of Hypoxic Conditioning Translational Medicine, Xuanwu Hospital, Capital Medical University, Beijing 100053, China; rench@xwhosp.org; 9Department of Neurosurgery, The First Affiliated Hospital of Baotou Medical College, Baotou 014010, China

**Keywords:** MALAT1, hypoxic/ischemic injury, neuronal cell, protection

## Abstract

Hypoxic and ischemic brain injury can cause neurological disability and mortality, and has become a serious public health problem worldwide. Long-chain non-coding RNAs are involved in the regulation of many diseases. Metastasis-related lung adenocarcinoma transcript 1 (MALAT1) is a type of long non-coding RNA (lncRNA), known as long intergenic non-coding RNA (lincRNA), and is highly abundant in the nervous system. The enrichment of MALAT1 in the brain indicates that it may be associated with important functions in pathophysiological processes. Accordingly, the role of MALAT1 in neuronal cell hypoxic/ischemic injury has been gradually discovered over recent years. In this article, we summarize recent research regarding the neuroprotective molecular mechanism of MALAT1 and its regulation of pathophysiological processes of brain hypoxic/ischemic injury. MALAT1 may function as a regulator through interaction with proteins or RNAs to perform its role, and may therefore serve as a therapeutic target in cerebral hypoxia/ischemia.

## 1. Introduction

Stroke, with a high disability rate, remains a leading cause of death in the world [1]. Ischemic stroke accounts for more than 80% of all kinds of stroke and is one of the most common manifestations of hypoxic and ischemic brain injury seen in clinics [2,3]. Ischemic stroke is generally caused by arterial embolization or the thrombotic occlusion of the brain. The brain is susceptible to hypoxic and ischemic injury due to its high demand for oxygen and glucose [4]; therefore, there is an urgent need to explore the molecular mechanisms of hypoxic and ischemic injury to search for targeted therapeutics for neuroprotection.

Non-coding RNAs (ncRNAs) are a class of functional RNAs that have been reported to play a role in brain hypoxic/ischemic injury in a post-transcriptional manner, through the regulation of microRNA and circular RNA, and may be used as stroke diagnostic biomarkers and as therapeutic targets [5,6]. Among ncRNAs, long non-coding RNAs (lncRNAs) are the most abundant class, consisting of transcripts >200 nucleotides long without a protein-coding function [7]. The MALAT1 long intergenic non-coding RNA (lincRNA) was first identified using subtractive hybridization and has been shown to be significantly associated with metastasis in non-small cell lung cancer [8]. MALAT1 is actively involved in various physiologic processes, including alternative splicing, epigenetic modification of gene expression, and synapse formation [9]. It has been proven that the damage induced by hypoxia/ischemia involves a series of reactions, including inflammation, apoptosis, and autophagy [10]. An increase in MALAT1 expression has been associated with decreased susceptibility of the brain to ischemic stroke through the promotion of angiogenesis, inhibition of apoptosis, as well as inflammation and regulation of autophagy [11]. It has been proposed that MALAT1 may have a function in the maintenance of large and complex nervous systems under particular conditions [12].

In this article, we reviewed the progress in the research on the molecular mechanisms of MALAT1 in the context of neuroprotection under hypoxic and ischemic conditions and potential therapeutic applications.

## 2. MALAT1 and Its Function in the Brain

MALAT1, also named NEAT2, is a 6.7–7 kb nuclear-resident lncRNA in humans and mice [10,13]. Further classified as an intergenic lncRNA, MALAT1 is synthesized by RNA polymerase Pol II. It has a single exon gene structure and it is enriched in the nucleus. While MALAT1 lacks a 5′ end cap structure, the 3′ end contains a triple helix structure that can protect it from leaving the nucleus and entering the cytoplasm to be degraded. Studies have also found that the triple helix structure may serve as a translation enhancer [14]. In addition, the 3′ end of MALAT1 can also bind to other proteins to serve a regulatory role. Furthermore, lncRNAs normally serve as genome modification complexes and target transcription factors, and can also affect mRNA expression and regulate gene stability through a sponge action [14]. MALAT1 is highly conserved, stable, and can control epigenetic gene regulation and splicing.

Interestingly, it has been observed that some proteins can interact with the promoter of MALAT1, thereby regulating the expression of MALAT1 and affecting nerve cells under hypoxia. It has been reported that the transcription factor KLF4 can bind to the promoter of lncRNA MALAT1, increasing MALAT1 expression and further protecting ischemic brain microvascular endothelial cells (BMECs) from apoptosis [15]. In addition, three hypoxic response elements have been found in the MALAT1 promoter. During hypoxia, HIF-1α and HIF-2α bind to the MALAT1 promoter, resulting in activation of the transcription of MALAT1 to increase expression [16,17]. Another study showed that p53 can be used as an inhibitor to bind to MALAT1, thereby inhibiting transcription [18].

MALAT1 is highly abundant in the brain and plays an important role in synapse density [19]. MALAT1 modulates synapse formation in neurons through the regulation of gene expression involved in synapse formation and/or maintenance. Moreover, knock-down/overexpression of MALAT1 in cultured neuron cells has been shown to result in a decrease/increase in synaptic density [19]. MALAT1 may constitute a lncRNA-mediated lncRNA maturation network in highly active brain regions to regulate the expression of genes involved in synaptogenesis [20]. MALAT1 plays a critical role in the early step of neuronal differentiation through activation of the ERK/MAPK signaling pathway [21].

## 3. MALAT1 Interacts with Proteins and RNAs under Hypoxic/Ischemic Conditions

### 3.1. MALAT1 Interacts with Proteins

MALAT1 can bind with different proteins to mediate a myriad of cellular processes, including alternative splicing and epigenetics, which may contribute to hypoxic/ischemic tolerance [22]. MALAT1 may regulate the stability and splicing of RNA through complementary base pairing or complex formation with RNA-binding proteins (RbPs), thereby regulating various cell activities. Brain ischemia and ischemia reperfusion induce the aggregation of transactive response DNA-binding protein (TDP)43, which contributes to neuronal injury [23]. It has been shown that MALAT1 can also bind to TDP43 and inhibit the cleavage of TDP43 into its active form, TDP35, thereby reducing the level of nuclear IRF3 in resting cells, maintaining immune homeostasis and protecting cells [24]. MALAT1 binding to TDP43 is expected to protect against hypoxic–ischemic damage at the cellular level.

It has been reported that MALAT1 can bind with proteins, such as splicing factor proline and glutamine rich (SFPQ), and release polypyrimidine tract binding protein 2 (PTBP2) to regulate runt-related transcription factor (2RUNX 2) mRNA levels, which may contribute to the hypoxic/ischemic process [25]. SFPQ is an RNA binding protein that can also work as a splicing factor to participate in mRNA splicing. The base sequence of its gene is rich in GC and is highly expressed in the hippocampus and cerebral cortex, where it can regulate the development and regeneration of nerves. In addition, SFPQ is down-regulated in neurodegenerative diseases and other neurological injury diseases, such as ischemic stroke [23]. Therefore, it can be speculated that MALAT1 protects from nerve damage caused by hypoxia/ischemia by combining with SFPQ by regulating mRNA splicing.

In addition to regulating alternative splicing, MALAT1 can also regulate gene transcription and epigenetic modification. MALAT1 may compete with DNA methyltransferase (DNMT) 1 and cell division cycle associated 7 (CDCA7) to bind to the E2F1 transcription factor, thereby interrupting inhibition of DNMT1 and the activation of CDCA7 by E2F1 [26]. In many cases, hypoxia/ischemia can cause DNA promoter methylation of certain genes, thereby inhibiting the expression of the gene and causing damage [27,28]. It could be predicted that MALAT1 may combine with DNMT to reduce the methylation of certain gene promoters under hypoxic conditions and promote their expression, resulting in nerve cell protection [26]. Therefore, MALAT1 may have a protective effect on nerve cells through epigenetics.

It can be concluded that the interaction of MALAT1 with different proteins may regulate different processes, such as alternative splicing, epigenetics and gene transcription. These processes may contribute to the protection of cells against hypoxic damage, depending on the complex containing MALAT1.

### 3.2. MALAT1 Interacts with RNA

MALAT1 can act as a sponge for some kinds of microRNAs to regulate their level and protect neuronal cells from hypoxic/ischemic injury. MALAT1 can promote the expression of NLR family pyrin domain containing 3 (NLRP3) by down-regulating the miR-224-5p level by acting as a competitive endogenous RNA. It has been reported that miR-224-5p reduces microglia inflammatory activation by regulating the expression of NLRP3, which ultimately affects the NLRP3/IL-1β pathway in the hippocampus [29]. It has been shown that MALAT1 can indirectly regulate inflammation through miR-224-5p to protect the nervous system from hypoxic injury. In addition, MALAT1 can be used as an endogenous sponge to downregulate the abundance of miR-26b by directly binding to miR-26b and promoting the survival of BMECs under oxygen–glucose deprivation/reperfusion (OGD/R) conditions [9]. MALAT1 interacts with miR-26b and upregulates ULK2 expression, which in turn suppresses neuronal death [30]. These results indicate that MALAT1, through its actions as an endogenous sponge, can have a protective effect against hypoxic damage through its influence on microRNA.

In addition to being an endogenous sponge, directly binding to microRNA, MALAT1 can also protect nerve cells from hypoxic or ischemic injury through indirect regulation. MALAT1 increases the expression of vascular endothelial growth factor-A and angiopoietin-like protein 2 by targeting miR-145 to promote the proliferation and angiogenesis of bone marrow mesenchymal stem cells under OGD/R conditions. This process suggests that MALAT1 can modulate blood vessel proliferation under hypoxic conditions and benefit anti-hypoxic/anti-ischemic injury [31,32].

In short, MALAT1 can perform a regulatory role by interacting with proteins or RNAs that have neuroprotection effects.

## 4. The Neuroprotective Role of lncRNA MALAT1 in Hypoxia/Ischemia

MALAT1 may be a potential target for hypoxic/ischemic brain injury treatment. Studies have shown that MALAT1 can protect nerves by inhibiting cell inflammation and apoptosis, regulating cell autophagy, promoting angiogenesis, protecting the blood–brain barrier (BBB), and improving cognitive function.

### 4.1. MALAT1 Improves Cognitive Function

Cognitive dysfunction is an important factor in the pathogenesis of nerve damage, and can lead to detrimental effects on behavior [33]. The main reason for this may be the selective loss of neurons in the hippocampus and cortex. Therefore, improving cognitive function is important for the treatment of hypoxic/ischemic injury. Zhang et al demonstrated that genetic deletion of the *MALAT1* gene presented a larger brain infarct size and a worse neurological score in mice after MCAO compared to WT controls [34]. Shang et al. found that spatial learning and memory were improved in ischemia-reperfusion injury mice, in which MALAT1 was overexpressed by a lentiviral vector [35]. This proves that MALAT1 can improve cognitive dysfunction after hypoxia/ischemia.

### 4.2. MALAT1 Protects the Blood–Brain Barrier (BBB)

The destruction of the BBB is an important process of secondary damage to brain nerves. The BBB is a semi-permeable barrier formed by tightly connected endothelial cells, astrocytes, pericytes, and basement membranes [36]. It can separate blood from the extracellular fluid in the brain to limit circulating white blood cells, blood-derived molecules, and toxic substances from entering the brain. Brain ischemia can cause damage to endothelial cells, leading to destruction of the BBB and increasing its permeability. It is well known that MALAT1 is highly expressed in endothelial cells and maintains the normal function of cerebral blood vessels [37,38]. The major protective effect of lncRNA MALAT1 on the BBB depends on promoting the survival of BMECs, which are a source of endothelial cells under OGD/R conditions [9]. At the same time, the up-regulation of MALAT1 can increase microvascular integrity in ischemic stroke [39]. There are relatively few studies on the direct protection of the BBB by MALAT1; however, Ma et al. showed that MALAT1 played a role in BBB permeability through microRNA [40]. Therefore, further research is needed to explore the underlying mechanism of MALAT1 in the BBB.

### 4.3. MALAT1 Promotes Angiogenesis

Angiogenesis is beneficial for brain function after the nervous system is injured. It is speculated that lncRNA MALAT1 promotes angiogenesis to alleviate nerve damage in stroke. Salehi et al. showed that the increase in vascular growth factor (VEGF) after nerve injury can promote angiogenesis and nerve function recovery [41]. Cerebral angiogenesis can increase blood flow in blood vessels, which allows oxygen and nutrients to be transported to hypoxic tissues. In addition to up-regulating VEGF, lncRNA MALAT1 can also induce ANGPT2 by targeting miR-145 to promote the angiogenesis of BMECs under hypoxic and sugar-deprivation conditions [31]. MALAT1 may regulate angiogenesis through the 15-LOX1/STAT3 signal pathway, which may be a key target for the treatment of hypoxic injury [42]. MALAT1 can protect the formation of BMECs under OGD/R by interacting with the miR-205-5p/VEGFA pathway [43]. At the same time, the increased level of lncRNA MALAT protects endothelial cells from ischemic injury after I/R or OGD/R treatment [43]. This process promotes the autophagy and survival of BMECs by absorbing miR-26b and upregulating the expression of ULK2 [9]. Therefore, lncRNA MALAT1 can defend against the destruction of the BBB and protect BMECs from OGD/R damage through angiogenesis.

### 4.4. MALAT1 Affects Autophagy

Autophagy is a process of cell self-decomposition and it generally occurs after cells are damaged [44]. Autophagy plays an important role in maintaining the metabolism necessary for the survival of cells in the absence of nutrients or other stresses [45]. Autophagy-associated lncRNAs can promote the development of neurological diseases or slow their progression [46]. MALAT1 can regulate autophagy in ischemic stroke through sponging miRNA and regulating their effects on autophagy-related factors [46]. Shi et al found that MALAT1 can affect autophagy to protect neurons from hypoxic/ischemic damage. Autophagy exists at low levels under normal conditions and excessive autophagy can lead to massive loss of primary cortical neurons [47]. However, moderate autophagy is considered beneficial and can play a protective role against varying insults to the brain [48]. The exact role of autophagy in the pathogenesis of ischemic stroke is controversial due to its “double-edged sword” effect on cells [49]. MALAT1 is positively correlated with the degree of autophagy. MALAT1 mainly regulates autophagy through interaction with microRNA. MALAT1 lncRNA up-regulates the expression of SIRT1 by binding to miR-200c-3p, a kind of microRNA that is related to autophagy, and protects BMECs from oxygen- and glucose-deprivation damage [50].

### 4.5. MALAT1 Has Anti-Inflammatory Effects

Inflammation plays an important role in secondary damage caused by nervous system insults. In normal cells, inflammation is generally a cellular and molecular process, the purpose of which is to enable cells to fight against invading pathogens and restore the health of cells and tissues. However, in pathological brain injury, excessive inflammation can aggravate brain injury and lead to brain edema, BBB destruction, and the release of inflammatory factors. The pro-inflammatory factors E-selectin, MCP-1, and IL-6 are increased in ischemic stroke models [51]. Neuroinflammation plays an important role in the pathological process of stroke and impairs nerve tissue and cells [52]. While MALAT1 has been reported to be downregulated and primarily play an anti-inflammatory role in patients with cerebrovascular disease [53], MALAT1 expression is negatively correlated with NIHSS (National Institutes of Health Stroke Scale) score and pro-inflammatory factor expression (including CRP, TNF-α, IL-6, IL-8, and IL-22) in acute ischemic stroke (AIS) patients. AIS patients with a higher MALAT1 expression have lower NIHSS and pro-inflammatory factor expressions [54]. In addition, Silencing lncRNA MALAT1 further increases pro-inflammatory factors, which indicates that MALAT1 has anti-inflammatory effects [34]. Moreover, MALAT1 has been proposed as a key player in regulating inflammation through NF-κB and AKT pathways in various neuropathologies of cerebral ischemic injury [52]. Therefore, the effects of MALAT1 on inflammation are, not only regulated by transcription factors, but also by ncRNA at the post-transcriptional level. It should be noted that the NF-κB pathway may be an important target for MALAT1 to regulate inflammation and protect nerve cells.

### 4.6. MALAT1 and Apoptosis

Reducing apoptosis and proteins with apoptosis-related functions can decrease brain damage under ischemic/hypoxic conditions. The overexpression of lncRNA MALAT1 can reduce cell apoptosis in brain I/R injury by activating PI3K and phosphorylated AKT in the PI3K/AKT pathway in vascular endothelial cells [13,55]. Yang et al showed that the anti-apoptosis role of MALAT1 in endothelial cells may depend on the Kruppel-like family of transcription factor 4 (KLF4), which is regulated by MALAT1 at the transcription level [15]. The in vitro and in vivo roles of MALAT1 showed differing effects on apoptosis and should be further researched. It is worth noting that MALAT1 plays different roles in different pathways, so its effect on neuronal apoptosis may be different in vivo and in vitro.

The mitochondria play key roles in activating apoptosis in mammalian cells and are regulated by Bcl-2 family proteins, including pro-apoptotic and anti-apoptotic proteins [56,57]. Wang et al reported that mitochondrial inner membrane cristae in hungry cells increased, and apoptosis was reduced or disappeared from the inner mitochondrial membrane crest [58]. Overexpression of MALAT1 can increase the Bcl-2 level and relieve apoptosis of neurocytes under OGD/R conditions [59]. Therefore, it is speculated that MALAT1 is involved in mitochondrial apoptosis and whether it interacts with certain substances during mitochondrial activity to protect against adverse reactions, such as hypoxia/ischemia.

## 5. MALAT1-Related Histone Modification Affected by Hypoxia/Ischemia

Several pieces of evidence indicate that hypoxic injury is closely related to histone modification [60]. The modification of histones prevents regions of the chromatin from contacting one another and changes the interaction of non-histone proteins with chromatin [61]. The most common modifications of histones are acetylation and methylation. It has been pointed out that MALAT1 can regulate gene transcription by interacting with histone modification enzymes and transcription factors [62]. The upregulation of oxygen-dependent KDMs may demethylate methylated lysine residues [63]. In addition, histone deacetylase (HDAC) 3-dependent deacetylation under acute hypoxic conditions can alter functional balance. EZH2-interacting MALAT1 across different tissues, including the brain, catalyzes histone methylation at H3 Lys27 and participates in diverse cellular processes. Although there has been no report about histone modification and MALAT1 in terms of neuroprotection under hypoxia/ischemia conditions, it can be proposed that MALAT1 may play a neuroprotective role through histone acetylation or methylation.

## 6. Concluding Remarks

Taken together, MALAT1 plays a vital role in hypoxic/ischemic tolerance in the brain by modulating cellular inflammation processes, apoptosis, and autophagy (Figure 1). lncRNA MALAT1, localized to nuclear bodies known as nuclear speckles, can work as regulators to maintain homeostasis in neurons under hypoxic/ischemic conditions. A change in MALAT1 expression affects BBB permeability, angiogenesis, and even cognitive functions. The interaction between MALAT1–RNA and MALAT1–protein is a key point in regulating its neuroprotective function in the brain under hypoxic/ischemic conditions.

However, the function of MALAT1 has not yet been clearly characterized, and its neuroprotective effects in the pathological process of ischemic stroke are not totally understood. Thus, further efforts should be made to investigate the neuroprotective molecular mechanisms of MALAT1. With a better understanding of the function of MALAT1, greater progress in alleviating hypoxic or ischemic damage can be achieved. These efforts will provide novel insights into MALAT1, which works as a potential therapeutic target for hypoxic and ischemic brain injury.

## Figures and Tables

**Figure 1 biomolecules-12-00146-f001:**
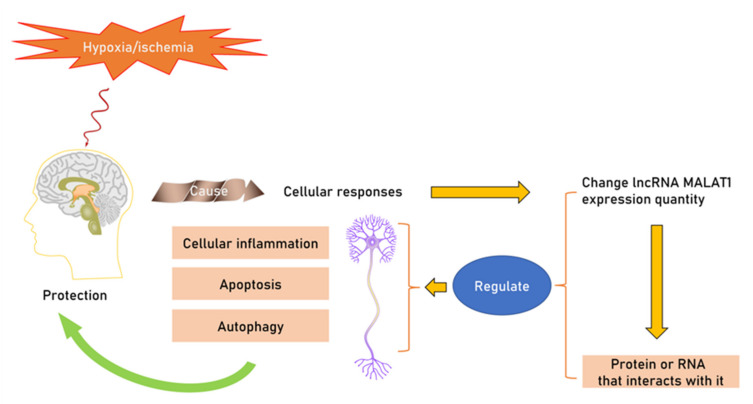
The neuroprotective function of MALAT1 under hypoxic/ischemic conditions. MALAT1 functions as a regulator during the transcriptional and post-transcriptional process through interaction with proteins or RNAs. In addition, neuroprotective mechanisms including inflammation, apoptosis and autophagy are shown.

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
