# Peer review of "The Role of the lncRNA MALAT1 in Neuroprotection against Hypoxic/Ischemic Injury"

_biomolecules, 2022, doi:10.3390/biom12010146_

Round 1

Reviewer 1 Report

This is a nicely written review describing the function and regulation of IncRNA MALAT1 in CNS, as well as its roles in neuroprotection under the condition of brain hypoxia-ischemia. Overall, this is a clear and strong presentation, and this reviewer finds no major weakness.

Minor comments are listed below.

  1. A list of abbreviations should be provided at the end of the manuscript.
  2. Title 4 and title 5 are overlapping, what about change title 4 to: MALAT1 interacts with proteins and RNAs under hypoxic/ischemic conditions
  3. The first paragraph of section 5 (line 205-218) is repetitive and lengthy, could be more concise to only introduce the content of this section.
  4. References are needed for statements described on lines 85-87
  5. The quality and resolution of Figure 1 is not high enough, and the legend should be more concise and clearer.
  6. Some grammatical errors, for example:
  • Line 29-30: should be hypoxic/ischemic injury
  • Line 34: extra “during”
  • Line 58: no “the” before susceptibility
  • Line 67: and is enriched in the nuclei
  • Line 148-149: use the abbreviation TDP43 when appeared the second time
  • Line 202: perform a regulatory role
  • Line 203: have neuroprotective effects
  • Line 234: omit “in” before blood flow
  • Line 312: should be further researched

Author Response

Comments 1, “1. A list of abbreviations should be provided at the end of the manuscript”.

Response:  According to the reviewer’s advice, we had added a list of abbreviation at the end of manuscript.

Comments 2, “Title 4 and title 5 are overlapping, what about change title 4 to: MALAT1 interacts with proteins and RNAs under hypoxic/ischemic conditions"

Response:  According to the reviewer’s advice, we had change title 4 to: MALAT1 interacts with proteins and RNAs under hypoxic/ischemic conditions. Please refer to line 95 for the detailed changes.

Comments 3, “The first paragraph of section 5 (line 205-218) is repetitive and lengthy, could be more concise to only introduce the content of this section”  

Response:  According to the reviewer’s suggestion, we had deleted a sentence in this section to make more concise.

Comments 4, “References are needed for statements described on lines 85-87”  

Response:  According to the reviewer’s suggestion, we had added references. Please refer to line 84 for the detailed changes.

Comments 5, “   The quality and resolution of Figure 1 is not high enough, and the legend should be more concise and clearer.”  

Response:  According to the reviewer’s suggestion, we have replaced it with a higher resolution version.

Comments 6, “   Some grammatical errors, for example:

  • Line 29-30: should be hypoxic/ischemic injury
  • Line 34: extra “during”
  • Line 58: no “the” before susceptibility
  • Line 67: and is enriched in the nuclei
  • Line 148-149: use the abbreviation TDP43 when appeared the second time
  • Line 202: perform a regulatory role
  • Line 203: have neuroprotective effects
  • Line 234: omit “in” before blood flow
  • Line 312: should be further researched.”

Response:  According to the reviewer’s advice, we had corrected all the errors we can find.

Reviewer 2 Report

This manuscript presents a review of the literature regarding the role(s) of the lncRNA MALAT1 in hypoxia and ischemia. The authors propose that, while MALAT1 plays no perceptible role in normal brain development or function, it may play a role in neuropathology such as brain hypoxia and ischemia. Although this reviewer applauds the effort of compiling a comprehensive review, there are several concerns with the manuscript as currently written, including a few that are more significant than others:

  1. The overall structure of the manuscript does not flow well and should be rearranged and made more concise. For instance, section 2 ends with a transition sentence speaking to the role of MALAT1 in hypoxia and ischemia, but then section 3 is on epigenetics and it’s not until section 4 that the authors discuss hypoxia-ischemia. Several of the paragraphs are also repetitive from other parts of the manuscript (e.g. the first paragraph of section 6, the first paragraph of section 4, and the first 4 sentences of section 5, which could all be removed), go into too much detail that is not directly related to this topic (e.g. lines 75-78, lines 109-110, lines 199-201, lines 317-320, and lines 359-360, which could all be removed), or do not even mention MALAT1 (e.g. the first paragraph of section 5.6 and the second paragraph of section 6, both of which could be removed). There is also a paragraph at the very end of the manuscript (starting with “when the body…”) that seems spurious. If it’s part of the conclusion, it is repetitive of what is already stated in the first concluding paragraph, and if it’s not part of the conclusion, it’s not clear why it is there.
  2. The authors need to be more careful and attentive to the different pathologies that are discussed and the considerable different in pathophysiology between them. As currently written, the authors treat stroke, obstructive sleep apnea, and “heart disease” as all having the same effect on the brain, which is not the case. OSA and heart disease are both chronic and primarily (or solely in the case of OSA) hypoxia, whereas stroke is acute and combined hypoxia-ischemia. If the authors choose to include MALAT1 data from both acute and chronic as well as hypoxia and hypoxia-ischemia, they need to be much more transparent about which studies assess each (perhaps separating them into different sections of the manuscript) and clearly differentiate them. If the authors are looking for other stroke-like pathologies, cardiac arrest would be a much more analagous process than OSA or heart disease.
  3. The authors should stick with consistent terminology throughout the manuscript. They often switch back and forth between “ischemia/hypoxia” and “ischemia and hypoxia” and “hypoxia/ischemia,” and in the title they spell “ischaemic” different from the rest of the manuscript. Similarly, the authors use “OGD” in some places and “OGD/R” and “hypoxic and sugar deprivation” in others. They also use hyphens in some of the HIF and not others.
  4. Lines 96-98 could easily undermine this entire manuscript as it is currently written. The authors need to make it much clearer why they think that MALAT1 is important despite any clear association with normal brain development and function. Consider moving this sentence to the introduction and make a stronger case for why the data from reference 18 do not mean that MALAT1 is completely unimportant in the brain (which this sentence in its current position could easily be read as meaning).
  5. There are several leaps in logic that are too much for a review manuscript. For example, at the end of section 3, the authors propose that, because m(6)a modification of Alkbh5/Fto is important in cerebral ischemia, that m(6)a modification of MALAT1 may also be (though the authors provide no references for whether m(6)a modification of MALAT1 even occurs in the brain, let alone is important in cerebral ischemia). Similarly, section 4.2 starts off with the authors stating that MALAT1 can downregulate miR-224 and miR-224 can alter microglial inflammation, and since OSA includes microglial inflammation, MALAT1 could provide neuroprotection for OSA hypoxia. They do not show, however, that MALAT1 affects microglial inflammation itself in any pathology, let alone OSA. Overall, it seems like the authors are trying to take a very small amount of data surrounding MALAT1 and infusing unrelated and minimally related data in order to try to make it sound more important than it is. Perhaps the authors could consider revising this into a letter or short communication and only including the few relevant reverences that they have and removing the spurious information.

In addition, there are several more minor concerns that the authors should consider addressing:

  • The abstract should have consistent tense (the last sentence is present tense while the rest is past tense)
  • BBB does not need to be abbreviated in the abstract as it is not used more than once
  • Lines 39-40: this gets to the issue of #2 above. Stroke is hypoxia and ischemia, not “hypoxia or ischemia.” Additionally, line 43 suggests that OSA and heard disease cause “hypoxic and ischemic” brain injury, which is inaccurate. These should both be corrected for accuracy.
  • In line 57, “cognitive impairment” is not a “reaction” and so should be removed from the list, or the sentence should be restructured.
  • Lines 92-94 appear misplaced as they do not fit with the sentences before or after them. Consider moving or removing.
  • Lines 114-116: unpublished data are not ideal for review articles since they cannot be verified by the readers. Consider removing this sentence or provide a published reference.
  • Lines 168-170: This sentence needs a reference
  • Line 183: OSA has already been defined so does not need to be defined again
  • Line 187: BMEC needs to be defined
  • Lines 187-191: The authors spend much of the manuscript describing how MALAT1 is neuroprotective, and then in this sentence they say that it is probably injurious and then just move on without further explanation. They must provide some explanation for why these authors found MALAT1 to be injurious when others found it to be protective.
  • The authors need to revise the titles of section 4 and/or 5 to better differentiate them. As currently written, they are almost identical and it is unclear what the difference is between the two.
  • Section 5.1 is entitled “MALAT1 improves cognitive function” but the only data they present is on motor function. The authors need to either change the title or alter the content to fit the title.
  • Section 5.3 is two full paragraphs to describe one single pertinent study. This section should either be more concise or include more pertinent information. The first paragraph in this section should also be better referenced (currently has zero references).
  • Line 274: Reference 42 is in an unrelated pathology of Alzheimer’s; as such, this reference should be replaced with a more applicable one, or the sentence removed
  • Line 276: Similarly, reference 55 is in TBI (see comment from line 274 above)
  • Lines 286-288: This sentence needs a reference
  • Lines 307-311: The authors should make it clearer what their proposal is for why the two studies described are contradictory to each other. It is possible that they currently attempt to do this at the end of the paragraph, but if that is the case, the description should be moved to immediately following these sentences.
  • Line 311: This sentence is opinion and therefore should not have a reference after it (i.e. remove reference 10 from this line)
  • Line 315: How did the authors determine that “the mitochondrial apoptosis pathway is the most important apoptosis pathway”? I suspect there is not data to support this statement, so it should be re-worded for accuracy.
  • Lines 355-356: Remove these sentences discussing cardiomyocytes and osteosarcoma, as they are unrelated to the topic of this manuscript.
  • Part of the conclusion should include what the authors feel are the current gaps in the literature on this subject.
  • Figure 1: “Various responses” is too generic and should be better clarified

Author Response

Comments 1, “The overall structure of the manuscript does not flow well and should be rearranged and made more concise. For instance, section 2 ends with a transition sentence speaking to the role of MALAT1 in hypoxia and ischemia, but then section 3 is on epigenetics and it’s not until section 4 that the authors discuss hypoxia-ischemia. Several of the paragraphs are also repetitive from other parts of the manuscript (e.g. the first paragraph of section 6, the first paragraph of section 4, and the first 4 sentences of section 5, which could all be removed), go into too much detail that is not directly related to this topic (e.g. lines 75-78, lines 109-110, lines 199-201, lines 317-320, and lines 359-360, which could all be removed), or do not even mention MALAT1 (e.g. the first paragraph of section 5.6 and the second paragraph of section 6, both of which could be removed). There is also a paragraph at the very end of the manuscript (starting with “when the body…”) that seems spurious. If it’s part of the conclusion, it is repetitive of what is already stated in the first concluding paragraph, and if it’s not part of the conclusion, it’s not clear why it is there.”

Response:  According to the reviewer’s suggestion, we had rearranged the structure of this manuscript, and deleted section 2 and some sentences to make this review manuscript more concise and flow.

Comments 2, “The authors need to be more careful and attentive to the different pathologies that are discussed and the considerable different in pathophysiology between them. As currently written, the authors treat stroke, obstructive sleep apnea, and “heart disease” as all having the same effect on the brain, which is not the case. OSA and heart disease are both chronic and primarily (or solely in the case of OSA) hypoxia, whereas stroke is acute and combined hypoxia-ischemia. If the authors choose to include MALAT1 data from both acute and chronic as well as hypoxia and hypoxia-ischemia, they need to be much more transparent about which studies assess each (perhaps separating them into different sections of the manuscript) and clearly differentiate them. If the authors are looking for other stroke-like pathologies, cardiac arrest would be a much more analagous process than OSA or heart disease.”

Response:  Thank you for pointing this out and we deleted these sentences about OSA and heart disease in revised manuscript.

.

Comments 3, “The authors should stick with consistent terminology throughout the manuscript. They often switch back and forth between “ischemia/hypoxia” and “ischemia and hypoxia” and “hypoxia/ischemia,” and in the title they spell “ischaemic” different from the rest of the manuscript. Similarly, the authors use “OGD” in some places and “OGD/R” and “hypoxic and sugar deprivation” in others. They also use hyphens in some of the HIF and not others”

Response:  Thank you for pointing out this and we apologize for the mistakes. Accordingly, we had unified terminology throughout the manuscript.

Comments 4, “Lines 96-98 could easily undermine this entire manuscript as it is currently written. The authors need to make it much clearer why they think that MALAT1 is important despite any clear association with normal brain development and function. Consider moving this sentence to the introduction and make a stronger case for why the data from reference 18 do not mean that MALAT1 is completely unimportant in the brain (which this sentence in its current position could easily be read as meaning)”

Response: We totally agree on the point that the sentence can undermine this entire manuscript. According to the reviewer’s suggestions, we had removed it to “introduction” section and rewritten it. Please refer to line 60-61 for the detailed changes.

Comments 5, “There are several leaps in logic that are too much for a review manuscript. For example, at the end of section 3, the authors propose that, because m(6)a modification of Alkbh5/Fto is important in cerebral ischemia, that m(6)a modification of MALAT1 may also be (though the authors provide no references for whether m(6)a modification of MALAT1 even occurs in the brain, let alone is important in cerebral ischemia). Similarly, section 4.2 starts off with the authors stating that MALAT1 can downregulate miR-224 and miR-224 can alter microglial inflammation, and since OSA includes microglial inflammation, MALAT1 could provide neuroprotection for OSA hypoxia. They do not show, however, that MALAT1 affects microglial inflammation itself in any pathology, let alone OSA. Overall, it seems like the authors are trying to take a very small amount of data surrounding MALAT1 and infusing unrelated and minimally related data in order to try to make it sound more important than it is. Perhaps the authors could consider revising this into a letter or short communication and only including the few relevant reverences that they have and removing the spurious information.”

Response:  Thank you for the comments and we deleted section 3, as well as all sentences about OSA and heart diseases, and rearranged some sentences to make this review manuscript more logic.

Comments 6, “   The abstract should have consistent tense (the last sentence is present tense while the rest is past tense)”

Response:  According to the reviewer’s suggestion, the mistake has been corrected. We also read through the manuscript and hope we have corrected all the tense related mistakes. 

Comments 7, “BBB does not need to be abbreviated in the abstract as it is not used more than once”

Response: According to the reviewer’s suggestion, we had removed the abbreviation of BBB.

Comments 8, “   Lines 39-40: this gets to the issue of #2 above. Stroke is hypoxia and ischemia, not “hypoxia or ischemia.” Additionally, line 43 suggests that OSA and heard disease cause “hypoxic and ischemic” brain injury, which is inaccurate. These should both be corrected for accuracy”

Response: According to the reviewer’s suggestion, we had rearranged the issue of #2. Thank you for pointing this out that “stroke is hypoxia and ischemia” and we had corrected accordingly. At the same time, we had deleted the sentences about “OSA and heart disease”.

Comments 9, “   In line 57, “cognitive impairment” is not a “reaction” and so should be removed from the list, or the sentence should be restructured.”

Response:  According to the reviewer’s suggestion, the sentence about “cognitive impairment” has been removed.

Comments 10, “ Lines 92-94 appear misplaced as they do not fit with the sentences before or after them. Consider moving or removing.”

Response:  According to the reviewer’s suggestion, we had rearranged this sentence. Please refer to line 91-93 for the detailed changes.

Comments 11, “  Lines 114-116: unpublished data are not ideal for review articles since they cannot be verified by the readers. Consider removing this sentence or provide a published reference.”

Response:  According to the reviewer’s suggestion, this sentence has been removed. 

Comments 12, “ Lines 168-170: This sentence needs a reference”

Response:  According to the reviewer’s suggestion, we had added references. Please refer to line 95 for the detailed changes. 

Comments 13, “ Line 183: OSA has already been defined so does not need to be defined again”

Response:  We had removed all sentence about OSA.

Comments 14, “ Line 187: BMEC needs to be defined”

Response:  According to the reviewer’s suggestion, BMEC has been defined in revised version.

Comments 15, “ Lines 187-191: The authors spend much of the manuscript describing how MALAT1 is neuroprotective, and then in this sentence they say that it is probably injurious and then just move on without further explanation. They must provide some explanation for why these authors found MALAT1 to be injurious when others found it to be protective.”

Response:  According to the reviewer’s suggestion, we had removed this sentence and added another reference to support the neuroprotective role of MALAT1. Please refer to line144-145 for the detailed changes.

Comments 16, “ The authors need to revise the titles of section 4 and/or 5 to better differentiate them. As currently written, they are almost identical and it is unclear what the difference is between the two.”

Response:  Thanks for the comments and the title of section 4 has been changed to “MALAT1 interacts with proteins and RNAs under hypoxic/ischemic conditions”. Please refer to line 95 for the detailed changes.

Comments 17, “ Section 5.1 is entitled “MALAT1 improves cognitive function” but the only data they present is on motor function. The authors need to either change the title or alter the content to fit the title”

Response:  According to the reviewer’s suggestion, we had altered the contents to fit the title.  Please refer to line 167-171 for the detailed changes.

Comments 18, “ Section 5.3 is two full paragraphs to describe one single pertinent study. This section should either be more concise or include more pertinent information. The first paragraph in this section should also be better referenced (currently has zero references).”

Response:  Thank you for your comments. We merged the two full paragraphs into one by deleting some sentences. At the same time, some references were added into this section.

Comments 19, “ Line 274: Reference 42 is in an unrelated pathology of Alzheimer’s; as such, this reference should be replaced with a more applicable one, or the sentence removed”

Response:  According to the reviewer’s suggestion, we rewrote this sentence and changed one reference.  Please refer to line 229-236 for the detailed changes.

Comments 20, “ Line 276: Similarly, reference 55 is in TBI (see comment from line 274 above)”

Response:  According to the reviewer’s suggestion, we rewrote this sentence and changed one reference. Please refer to line 238-240 for the detailed changes.

Comments 21, “ Lines 286-288: This sentence needs a reference”

Response:  According to the reviewer’s suggestion, two references have been added. Please refer to line 181 for the detailed changes.

Comments 22, “ Lines 307-311: The authors should make it clearer what their proposal is for why the two studies described are contradictory to each other. It is possible that they currently attempt to do this at the end of the paragraph, but if that is the case, the description should be moved to immediately following these sentences.”

Response:  According to the reviewer’s suggestion, the description has been moved.

Comments 23, “ Line 311: This sentence is opinion and therefore should not have a reference after it (i.e. remove reference 10 from this line)”

Response:  According to the reviewer’s suggestion, the reference has been removed. 

Comments 24, “ Line 315: How did the authors determine that “the mitochondrial apoptosis pathway is the most important apoptosis pathway”? I suspect there is not data to support this statement, so it should be re-worded for accuracy.”

Response:  According to the reviewer’s suggestion, we had rewritten this sentence. Please refer to line 255-256 for the detailed changes.

Comments 25, “ Lines 355-356: Remove these sentences discussing cardiomyocytes and osteosarcoma, as they are unrelated to the topic of this manuscript.”

Response:  According to the reviewer’s suggestion, all sentences about cardiomyocytes and osteosarcoma have been removed.

Comments 26, “ Part of the conclusion should include what the authors feel are the current gaps in the literature on this subject.”

Response:  According to the reviewer’s suggestion, sentences about the current gaps in the literature on this subject have been added into the conclusion section. Please refer to line 279-293 for the detailed changes.

Comments 27, “ Figure 1: “Various responses” is too generic and should be better clarified.”

Response:  According to the reviewer’s suggestion, “Various responses” has been changed to “cellular responses”.

Round 2

Reviewer 2 Report

This reviewer appreciates the effort the authors have put into their revisions. A couple of issues still remain from the initial review, such as:

  • Abstract still has inconsistent tense – the authors changed all sentences instead of just the ones that were inconsistent (i.e. they changed the first sentences from present to past and the last sentence from past to present). All of the sentences should be present tense, so the authors should change all of the sentences back except the final sentence.
  • The spelling of ischemia is still inconsistent between the title/keywords and the remaining manuscript
  • Line 81, the Yang reference is not formatted.

Author Response

Comments 1, “Abstract still has inconsistent tense – the authors changed all sentences instead of just the ones that were inconsistent (i.e. they changed the first sentences from present to past and the last sentence from past to present). All of the sentences should be present tense, so the authors should change all of the sentences back except the final sentence.”.

Response:  According to the reviewer’s advice, the abstract has been rewritten. At the same time, the writing and grammar of this manuscript had been edited by an English editor from MDPI (english-38975).

Comments 2, “The spelling of ischemia is still inconsistent between the title/keywords and the remaining manuscript”.

Response:  According to the reviewer’s advice, we had corrected these errors.

Comments 3, “Line 81, the Yang reference is not formatted.”.

Response:  According to the reviewer’s advice, Yang reference has been formatted.